# CryoEM structure and Alphafold molecular modelling of a novel molluscan hemocyanin

**Gaia Pasqualetto**[1], **Andrew Mack**[1], **Emily Lewis**[1], **Ryan Cooper**[1], **Alistair Holland**[1], **Ufuk Borucu**[2], **Judith Mantell**[2], **Tom Davies**[3], **Miriam Weckener**[4], **Dan Clare**[5], **Tom Green**[6], **Pete Kille**[1], **Alex Muhlhozl**[7], **Mark T. Young**[1]*

1 School of Biosciences, Cardiff University, Cardiff, United Kingdom, 2 Faculty of Life Sciences, GW4 Facility for High-Resolution Electron Cryo-Microscopy, Wolfson Bioimaging Facility, University of Bristol, Bristol, United Kingdom, 3 School of Chemistry, Cardiff University, Cardiff, United Kingdom, 4 The Rosalind Franklin Institute, Structural Biology, Harwell Science Campus, Didcot, United Kingdom, 5 Electron Bioimaging Centre, Diamond Light Source, Harwell, United Kingdom, 6 Advanced Research Computing at Cardiff, Cardiff University, Cardiff, United Kingdom, 7 Mikota Ltd, Pembroke Dock, United Kingdom

* youngmt@cardiff.ac.uk

**Data Availability Statement:** CryoEM maps for SLH didecamer (EMD-16519) and tridecamer (EMD-16523) have been deposited with the Electron Microscopy Data Bank. CryoEM movies

## Abstract

Hemocyanins are multimeric oxygen transport proteins present in the blood of arthropods and molluscs, containing up to 8 oxygen-binding functional units per monomer. In molluscs, hemocyanins are assembled in decamer 'building blocks' formed of 5 dimer 'plates', routinely forming didecamer or higher-order assemblies with d5 or c5 symmetry. Here we describe the cryoEM structures of the didecamer (20-mer) and tridecamer (30-mer) forms of a novel hemocyanin from the slipper limpet *Crepidula fornicata* (SLH) at 7.0 and 4.7 Å resolution respectively. We show that two decamers assemble in a 'tail-tail' configuration, forming a partially capped cylinder, with an additional decamer adding on in 'head-tail' configuration to make the tridecamer. Analysis of SLH samples shows substantial heterogeneity, suggesting the presence of many higher-order multimers including tetra- and pentadecamers, formed by successive addition of decamers in head-tail configuration. Retrieval of sequence data for a full-length isoform of SLH enabled the use of Alphafold to produce a molecular model of SLH, which indicated the formation of dimer slabs with high similarity to those found in keyhole limpet hemocyanin. The fit of the molecular model to the cryoEM density was excellent, showing an overall structure where the final two functional units of the subunit (FU-g and FU-h) form the partial cap at one end of the decamer, and permitting analysis of the subunit interfaces governing the assembly of tail-tail and head-tail decamer interactions as well as potential sites for N-glycosylation. Our work contributes to the understanding of higher-order oligomer formation in molluscan hemocyanins and demonstrates the utility of Alphafold for building accurate structural models of large oligomeric proteins.

## Introduction

Hemocyanin is a large oligomeric oxygen transport protein present in the hemolymph of molluscs and arthropods. In arthropods, hemocyanin circulates in the hemolymph as hexamers

(EMPIAR-11554) have been deposited with EMPIAR. SLH sequence data have been deposited at Genbank with the accession codes OQ309180 (SLH1) and OQ309181 (SLH2 – truncated).

**Funding:** G.P., R.C., A.H., M.T.Y. and A.M. were supported by funding from Innovate UK (Technology Strategy Board grants 69001 (to G.P., M.T.Y. and A.M.), 36295 (to R.H., A.H, M.T.Y. and A.M.) and 14130 (to A.M.)). We would also like to thank the ERDF (European Regional Development Fund) and Wolfson Foundation for funding the CCI (Cardiff Catalysis Institute) Electron Microscopy Facility. The funders had no role in study design, data collection and analysis, decision to publish, or preparation of the manuscript.

**Competing interests:** The authors have declared that no competing interests exist.

[1]. In molluscs, hemocyanin is assembled in decamers: a single decamer for cephalopods (e.g. squids, octopuses, nautilus, cuttlefish) whilst di- or multi-decamers constitute the physiological assembly in gastropods (e.g. sea and terrestrial snail and slugs, limpets) [2].

Molluscan hemocyanins are classified into four structural types. Keyhole limpet-type hemocyanin is found in gastropods (e. g. the keyhole limpet (*Megathura crenulata*; KLH), *Rapana venosa*, etc), and has a hollow cylindrical shape formed by an assembly of 2 or more decamers (with the exception of *Bionphalaria glabrata* snail, which only shows single decamers [3]). Each decamer has D5 symmetry with 2 monomers forming a 'plate'. Each monomer is formed by 8 globular functional units named FU-a–FU-h, 6 of which (FU-a to FU-f) form the outer wall of the molecular assembly whilst FU-g and FU-h form the inner wall [4]. Mega-hemocyanin-type is the largest type (approx. 13.5 MDa), found in Cerithioid snails. It consists of a 3-decamer assembly made by 10 mega-subunits (core decamer formed by 12 functional units) and 20 regular subunits (formed by 8 functional units). The two 'regular' decamers stack one of each side of the mega-decamer [5,6]. Nautilus-type hemocyanin (approximately 3.5 MDa), found in nautilus and octopus, forms stable decamers and each monomer is formed by 7 FUs, lacking FU-h in the inner wall, resulting in a wider cavity of the decamer [7]. Squid-type hemocyanin is found in the decapodiformes order of cephalopods (cuttlefish, squids), also presenting as stable single decamers and lacking FU-h [8].

Two different isoforms of hemocyanin have been observed in KLH and hemocyanins from other gastropods (e.g. *Haliotis diversicolor*). The KLH isoform 1 structure has been solved by cryoEM at 9 Å resolution [4] and more recently at 6.5 Å resolution using scanning transmission electron microscopy [9]. Each decamer displays D5 symmetry and is formed by 5 plates, each formed by 2 monomers interacting asymmetrically. Each functional unit contains the copper-oxygen binding site, which has a highly conserved structure across all the different functional units, where two atoms of $Cu^{2+}$ are coordinated by 6 conserved histidine residues. KLH has also been reported to be able to form different aggregation states (didecamer or multidecamer assemblies) and this is reportedly influenced by divalent cation concentration ($Ca^{2+}$ and $Mg^{2+}$) [10]. Another characteristic of hemocyanins is their glycosylation [11], thought to allosterically contribute to the binding of oxygen molecules as well as stabilizing the assembly through inter-subunit interactions [12].

Different connectivities of the functional units within the asymmetric dimer have been proposed for KLH [4] and for *Haliotis diversicolor* hemocyanin [12]. For KLH, Gatsogiannis and Markl described one compact monomer and one 'extended' monomer, where FU-g and FU-h are extended relative to FU-a–FU-f [4]. For *Haliotis diversicolor* hemocyanin, Zhang *et al*. found that the two monomers forming the asymmetric dimer are more intertwined (effective 'domain swapping' of FU-e and FU-f), conferring stability to the plate dimer [12].

In order to assess the oligomeric formation of molluscan hemocyanin, and to investigate the architecture of the asymmetric dimer plate, we chose to study a novel molluscan hemocyanin, that of *Crepidula fornicata*, or slipper limpet using cryoEM. The slipper limpet is an invasive marine gastropod originally from North America, which has colonised European shores in the last decades. Slipper limpets are protandrous hermaphrodites, living in stacks of multiple individuals where the organism at the bottom is the only female. They reproduce twice a year and individuals reach sexual maturity in just 1–2 years. Their higher breeding rate, their adaptability to wide range of habitats and the lack of natural predators has caused them to outcompete the native species such as the native European oyster (*Ostrea edulis*) causing significant devastation of the shore beds [13,14].

We were able to resolve cryoEM structures of a didecamer (7.0 Å resolution) and a tridecamer (4.7 Å resolution) of slipper limpet hemocyanin (SLH), showing not only that SLH has a similar overall architecture to KLH, but also how single decamers are able to add to a 'core'

didecamer in mega-hemocyanin assembly. Furthermore, we were able to produce molecular models of individual monomers, dimer plates, didecamer and tridecamer structures using Alphafold, supporting the subunit arrangement proposed by Gatsogiannis and Markl [4], and providing molecular models of the subunit interfaces at the core of the didecamer and the didecamer-tridecamer junction. Our work provides 3D-structures for a novel hemocyanin molecule and exemplifies the power and utility of Alphafold for modelling large multimolecular protein assemblies.

## Results

### SLH forms heterogeneous populations of di-, tri-and multidecamers

Following stabilisation in Phase 2 buffer (50 mM Tris-HCl, 150 mM NaCl, 5 mM $MgCl_2$ and 5 mM $CaCl_2$), SLH was analysed by SDS-PAGE and gel filtration chromatography. SLH presented as a single band of approximately 350kDa with a similar molecular mass to KLH (Fig 1A), and a single copper-containing asymmetric peak at an elution volume of approximately 9 ml following separation on a Superose 6 10/300 GL column (Fig 1B). The low elution volume on Superose 6, coupled with the asymmetric peak, indicated the likely presence of more than one high-molecular mass species in the sample. Preliminary negative-stain transmission electron microscopy (Fig 1C) and cryo transmission electron microscopy (cryoEM) analysis (Fig 1D) confirmed the heterogeneous nature of SLH, with only top- and side-views observed in negative-stain, but with more tilted side-views observed in cryoEM. Particles corresponding to didecamers, tridecamers and higher-order oligomers were observed. In an attempt to improve particle homogeneity (or potentially separate different isoforms), the sample was subjected to ion exchange chromatography using a HiTrap SP FF column (S1A Fig), and samples were collected from both a large unbound fraction and a small peak eluted with increasing NaCl gradient. Gel filtration (S1B and S1C Fig) and SDS-PAGE (S1D Fig) of each fraction showed identical profiles, indicating a dynamic equilibrium that prevents substantial improvement in homogeneity. Negative-stain TEM analysis of each fraction (unseparated, anion exchange flow-through and salt elution) showed heterogeneity in each fraction (S2A Fig), and analysis of the particle lengths from representative micrographs showed a small increase in the relative proportion of didecamer-sized (approximately 35 nm) particles in the anion exchange flow-through (S2B Fig), so this sample was used for subsequent analysis.

### CryoEM structures of SLH didecamer and tridecamer indicate higher-order assembly by successive head-tail addition of decamers

SLH cryo-samples were prepared using the automated Chameleon system and imaged on an FEI Glacios instrument. The resulting dataset was analysed using Relion 3.1 [15]. Representative motion-corrected images are shown in S3A Fig, and substantial particle heterogeneity can be observed. Rounds of automated particle selection using Relion and crYOLO [16] were used to pick a total of 4149 particles for the tridecamer, and manual selection using Relion was used to pick a total of 1541 particles for the didecamer. Representative 2D class averages for the didecamer and tridecamer are shown in S3B and S3C Fig respectively. The 3D-structure of the di- and tridecamer were built using d5 and c5 symmetry, respectively (Fig 2A–2C). The final resolution of the didecamer was 7.0 Å (derived from a subset of 710 particles) and tridecamer 4.7 Å (derived from 4149 particles) (FSC = 0.143 criterion; S3D and S3E Fig). The overall structure of the SLH didecamer (Fig 2A–2C) closely resembles that of other KLH-type hemocyanins; a hollow cylinder (length 35 nm, diameter 32 nm) with a partial cap at either end, and didecamers arranged in a tail-tail configuration. The tridecamer (length 49 nm) can readily be

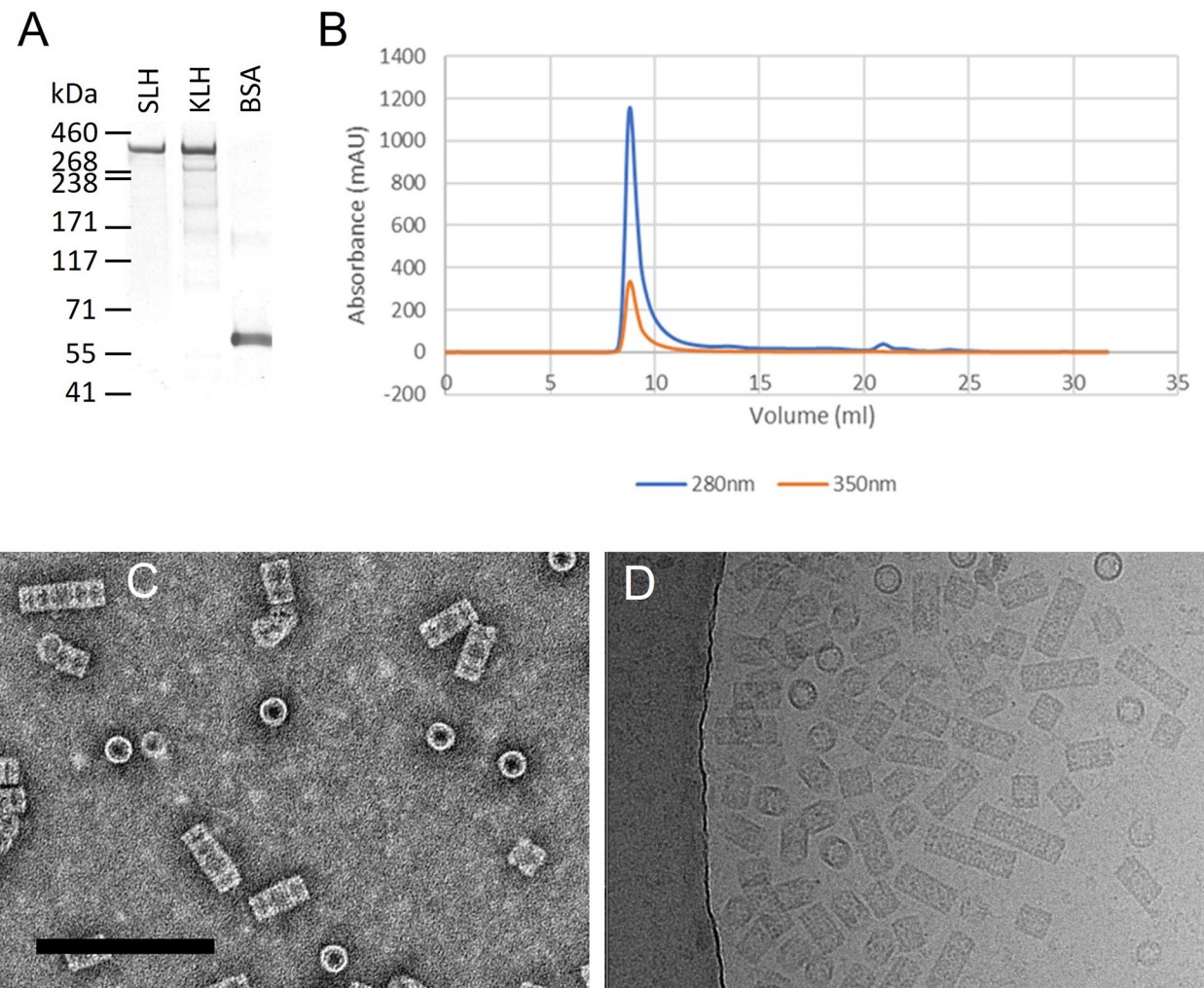

**Fig 1. Purification and EM analysis of slipper limpet hemocyanin.** A. SDS-PAGE analysis (3–8% Tris-acetate) of SLH. Composite image (individual lanes cropped from the same gel) of freshly isolated SLH, KLH (Merck) and BSA (Fraction V, Formedium). 1 μg protein loaded per lane. B. Gel filtration chromatography of SLH (Superose 6 10/300 column; 0.5 ml freshly isolated SLH (approx. 6 mg/ml) loaded) showing signals at 280 nm (blue; total protein) and 350 nm (orange; copper centre). C. Negative-stain TEM image (0.1 mg/ml SLH) showing top and side views. Scale bar 200 nm D. Cryo-TEM image (4 mg/ml SLH) showing particle heterogeneity.

superimposed on the didecamer (Fig 2A), revealing that it is composed of a didecamer with an additional decamer added onto one end in a head-tail configuration (see simplified representation in Fig 2B). Due to an insufficient number of particles, we were unable to determine the structure of any higher-order complexes. However, from our TEM analysis it is reasonable to deduce that the tail-tail didecamer constitutes the 'core' unit of any higher-order SLHs, and that 'mega'-hemocyanins are built by the successive addition of decamers at either end in head-tail configuration (S2C Fig).

## Sequence analysis and Alphafold modelling of SLH subunits

Analysis of publicly deposited transcriptome data (Bioproject PRJNA249058) using the sequence of KLH enabled assembly of an open reading frame (3424 amino acids) consistent with that of full-length SLH that we have termed SLH1 (S4 Fig). SLH 1 shares 54% identity at

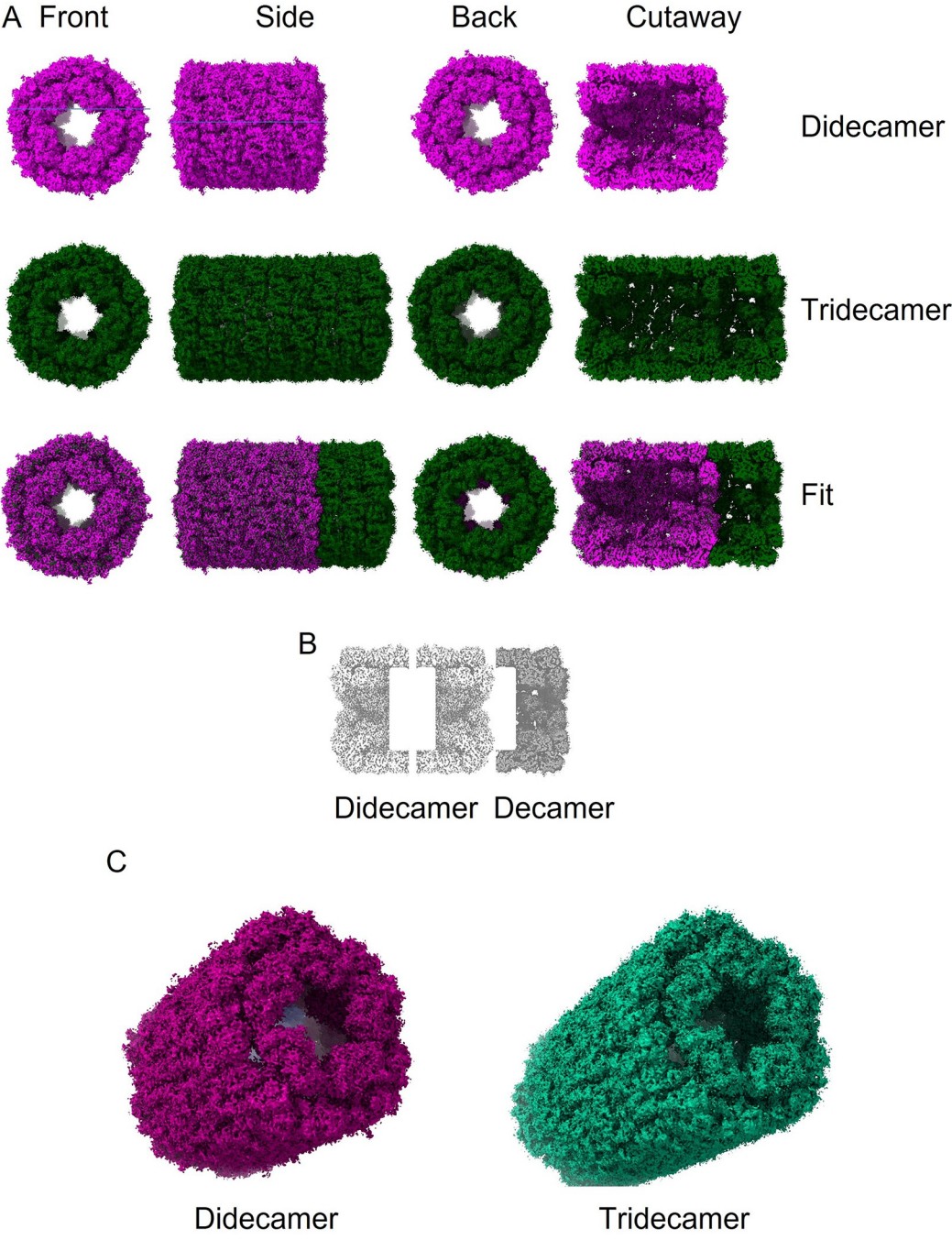

**Fig 2. CryoEM structures of slipper limpet hemocyanin didecamer and tridecamer.** A. Front, side, back and cutaway views of the 7 Å resolution didecamer (pink) and 4.7 Å resolution tridecamer (forest green) SLH cryoEM structures. B. Schematic view of how decamer units fit together in the tridecamer (the didecamer is formed by 'tail-tail' interactions and the additional decamer in the tridecamer forms a 'head-tail' interaction with one decamer in the didecamer). C. Oblique views of the didecamer (pink) and tridecamer (green) cryoEM structures.

the amino-acid level with KLH1 and 55% identity with KLH2 (KLH1 and KLH2 share 62% identity). Our analysis identified another potential SLH isoform (SLH2) that was 93% identical to SLH1 but truncated at amino acid 3277. The high degree of sequence identity between SLH1 and KLH1 suggests that the two proteins share the same overall structural fold, and we

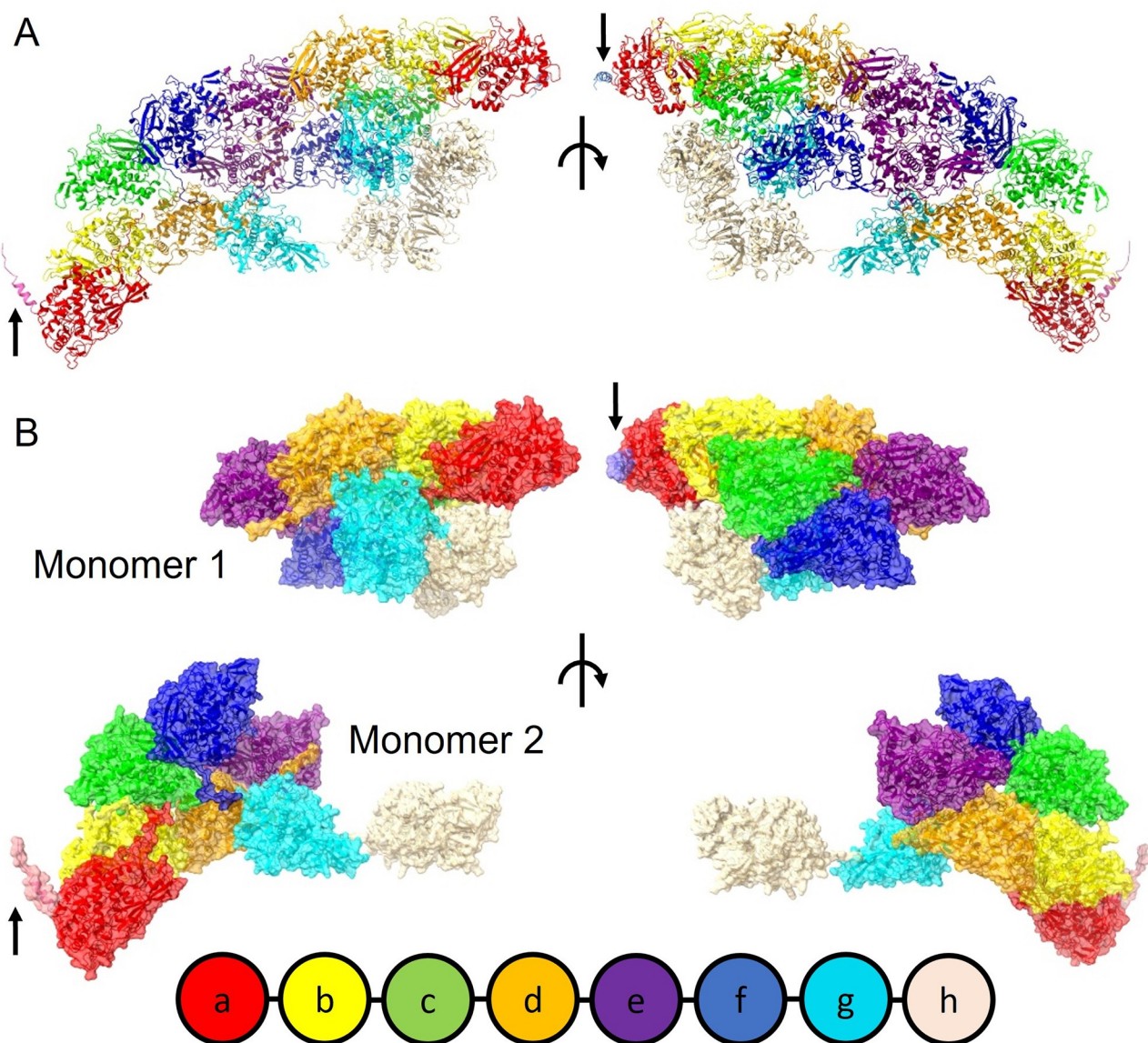

**Fig 3. Alphafold model of slipper limpet hemocyanin protomer.** A. Cartoon views (right-hand side rotated 180˚) of the SLH dimer (functional units colored according to the key at the bottom of the figure). B. Space-filling view (right-hand side rotated 180˚) of the separate SLH1 monomers in the model, showing the compact nature of one monomer (1) and the extended conformation of functional units g-h in the other monomer (2). Black arrows indicate the signal sequence (presumed to be cleaved in the mature secreted protein).

were able to map each of the 8 functional units found in KLH (FU-a–FU-h) on to the SLH1 sequence (S4 Fig; color scheme according to Fig 3).

Comprising 3424 amino acids, SLH1 is too large to model as a single chain using Alphafold. To circumvent this problem, we segmented the sequence into 6 overlapping fragments, predicted their structures and eventually superimposed them to create a full model of a single subunit. Two possible models emerged from our prediction, differing in the conformation of FU-g and FU-h (Fig 3B). We were able to fit these together to make a model of the dimer (Fig 3A), the repeating plate unit in the decamer structure. Our molecular model is strikingly similar to that of KLH [4], suggesting a similar subunit arrangement; a 'heterodimer' of homomers to make the repeating plate unit. In our models we observed a short sequence at the N-terminus

of each subunit (arrows in Fig 3); this was assumed to be signal sequence directing SLH biosynthesis to the lumen of the endoplasmic reticulum, which would be cleaved in the mature protein, so we removed it from the model for further analysis.

We were able to relate the dimers by five-fold symmetry to generate molecular models of the complete didecamer and tridecamer (Fig 4A); in these models the barrel structure is made up of FU-a–FU-f, and the partial caps at either end of the didecamer are made up of FU-g and FU-h. Using the models enabled us to segment the cryoEM maps by dimer 'slab' to show the relationship between the individual building blocks of the multimer (Fig 4B). The predicted and modelled structure was fitted into the cryoEM electron density map of the didecamer and tridecamer, respectively (Fig 5A), demonstrating its validity (at least at the level of the overall fold of the polypeptide chain). Close-up views highlighting one dimer in the cryoEM density for the didecamer and tridecamer (Fig 5B) demonstrates a very good fit at the single subunit level and a good fit of the model to the cryoEM density. The tridecamer model presents a high concentration of negative charge at surface of the cylinder walls, whereas the decamer interface is mostly hydrophobic with scattered positively charged residues (Fig 6). This indicates that tail-tail or head-tail interactions are energetically favoured whereas side-side interactions are not.

Because the association between subunits is tail-tail in the didecamer interface (monomer 2-monomer 2), and head-tail in the didecamer-tridecamer interface (monomer 1-monomer 2), we analysed the two interfaces in our molecular model of the tridecamer (Fig 7A and 7B). In the didecamer interface (Fig 7C), the subunit interaction appears to be predominantly mediated by FU-a, but in the didecamer-tridecamer interface (Fig 7D), it appears to be mediated by FU-a, FU-b and FU-c, as the interface appears to be rotated relative to the didecamer interface.

Molluscan hemocyanins are heavily N-glycosylated [11]. We used the NetNGlyc server [17] to analyse the presence of N-glycosylation acceptor sequences on SLH (S5A Fig). Of the potential 11 acceptor sequences, all were surface-exposed in the SLH molecular model, and 8 were predicted to be N-glycosylated by the NetNGlyc algorithm. A comparison to KLH, which is thought to contain 6 N-glycan chains per monomer and a total of 120 in the didecamer [4], revealed that 3 sites (Asn-410 in FU-a, Asn-1657 in FU-d and Asn-2493 in FU-f) are structurally conserved between the two proteins. We highlighted the Asn residues on our molecular model of the tridecamer (red for those conserved with KLH and blue for non-conserved) and fitted this to the tridecamer cryoEM density (S5B Fig). We did the same for one dimer plate of the didecamer cryoEM density (S5D Fig). We observed that some Asn sidechains were buried in the cryoEM density, and that some were not (S5A, S5C and S5D Fig). Each of the conserved Asn residues (410, 1657 and 2493) was associated with additional 'non-protein' cryoEM density, as were Asn-471 (FU-b), Asn-965 (FU-c) Asn-1045 (FU-c), Asn-1808 (FU-e) and Asn-3312 (FU-h). In our model the sidechains of Asn-3312 in FU-h face the centre of the SLH barrel structure. From our data it is not possible to unambiguously attribute additional density associated with the Asn sidechains to N-linked glycosylation, but we suggest that it is possible that the conserved sites between KLH and SLH are glycosylated, and potentially also another 5 non-conserved sites.

## Discussion

In this work we have determined the cryoEM structures of a novel molluscan hemocyanin, SLH, in both the didecamer and tridecamer form. We have generated a molecular model of the SLH protein using Alphafold, finding that it fits very well with our cryoEM structures. This has enabled us to determine relatively complete 3D-structures for these two structural isoforms

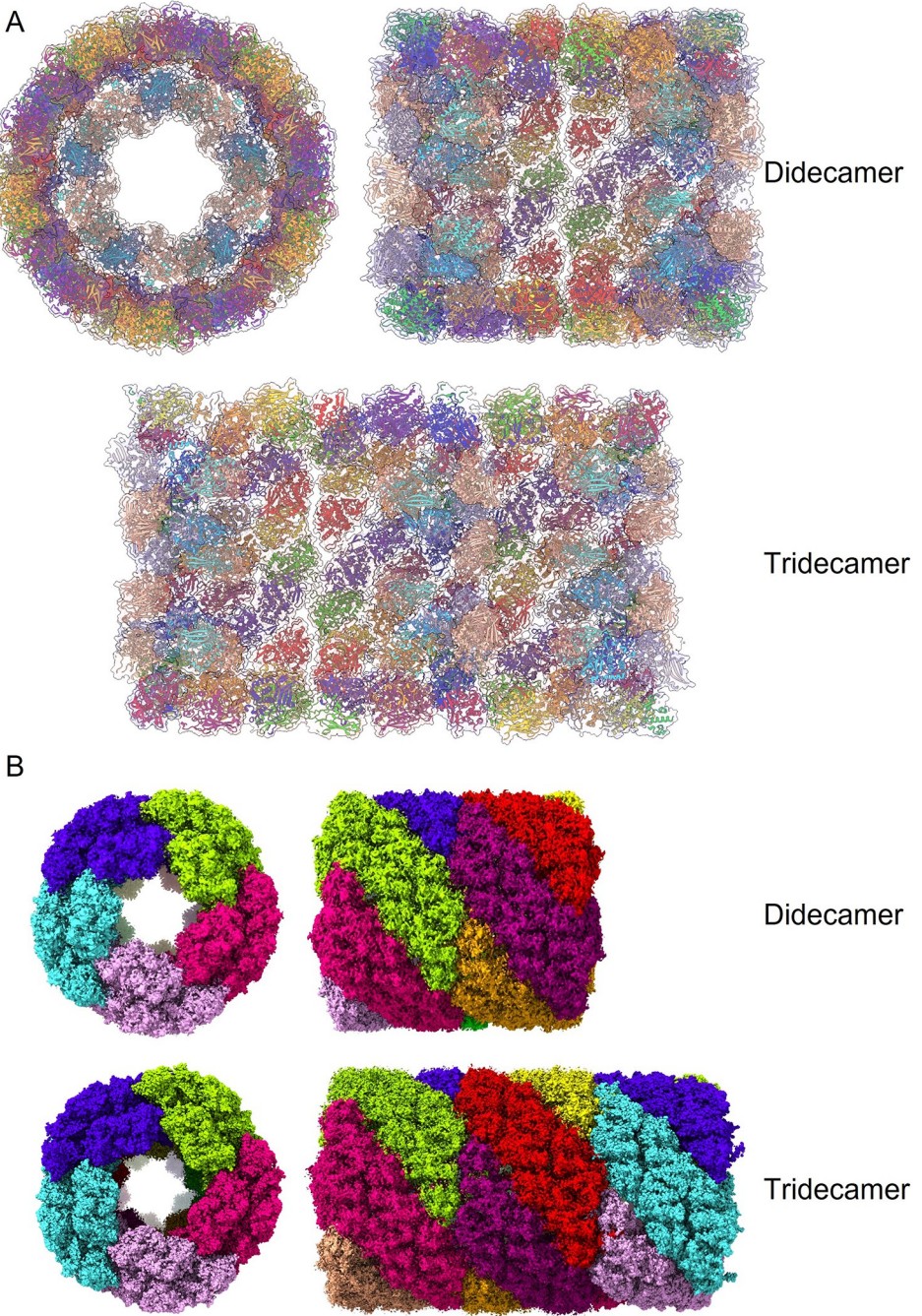

**Fig 4. Alphafold models of slipper limpet hemocyanin didecamer and tridecamer.** A. Molecular cartoon models of SLH didecamer and tridecamer (left, front view; right, cutaway side view), colored according to functional units, showing that functional units g and h are situated in the core of the 'barrel' structure. B. CryoEM structures of the didecamer and tridecamer with each dimer plate represented by a different color.

of SLH, and understand their molecular architecture in terms of the arrangement of functional units within each monomer, and how mega-hemocyanin structures can be built by adding decamers in a head-tail configuration onto 'core' didecamer structures.

One of the significant challenges we faced when attempting to determine the cryoEM 3D-structure of SLH was the sample heterogeneity (presence of different multimeric structures,

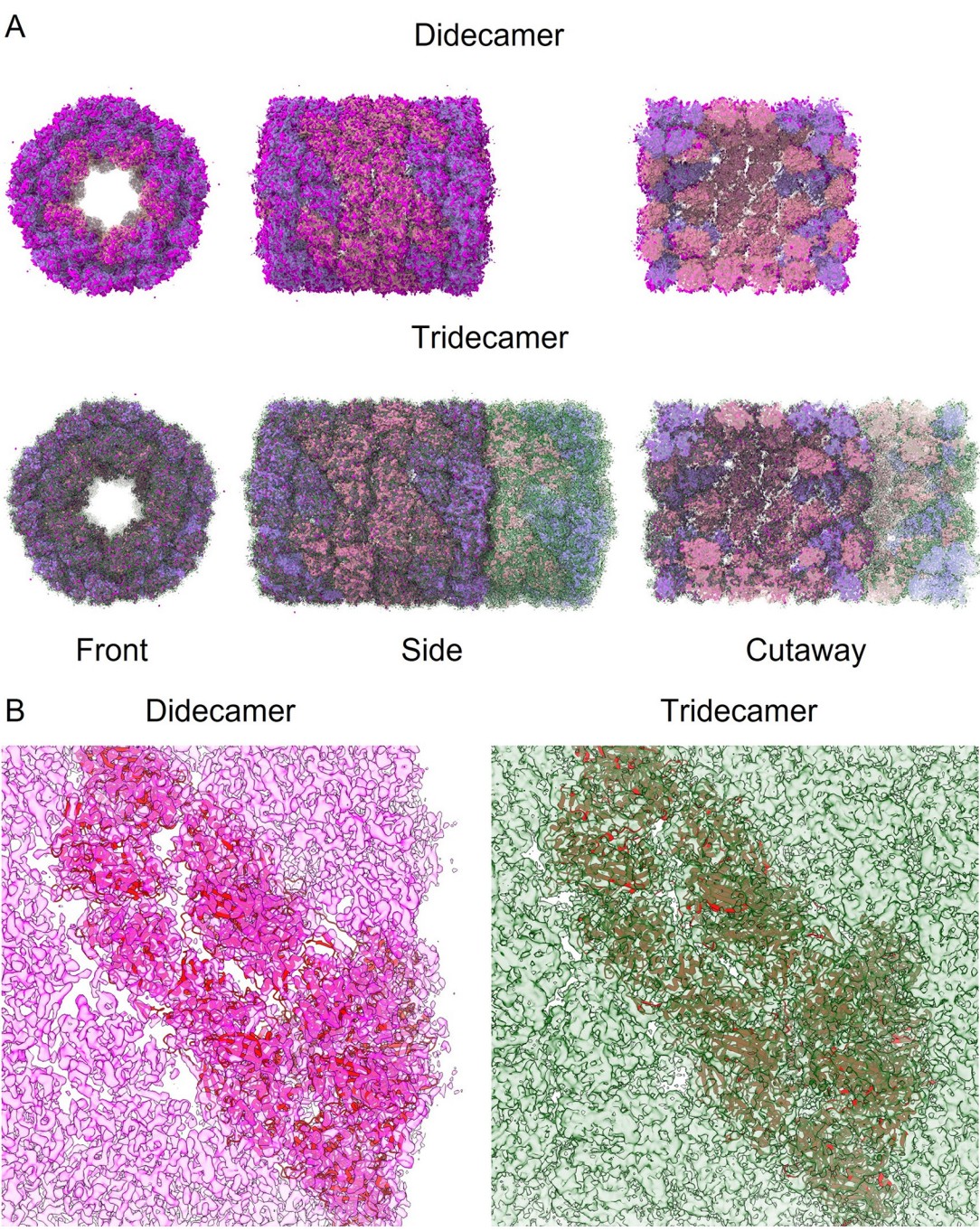

**Fig 5. Fitting of Alphafold models into slipper limpet hemocyanin cryoEM structures.** A. Front, side and cutaway views of the didecamer (pink) and tridecamer (forest green) cryoEM structures with alphafold space-filling models (monomer 1 light pink; monomer 2 light purple) fitted into them. B. Close-up of the alphafold model (one dimer; red ribbon) fitted to the cryoEM structures of the didecamer (pink) and tridecamer (green) showing the closeness of fit of the molecular model to the cryoEM volume.

and the predominance of larger multimers including tridecamers, tetradecamers and pentadecamers). This is commonly observed in hemocyanin preparations from *Haliotis* and keyhole limpet, but can be partially resolved by further purification, in particular ion exchange chromatography [10]. In the case of SLH, we were not able to substantially reduce sample

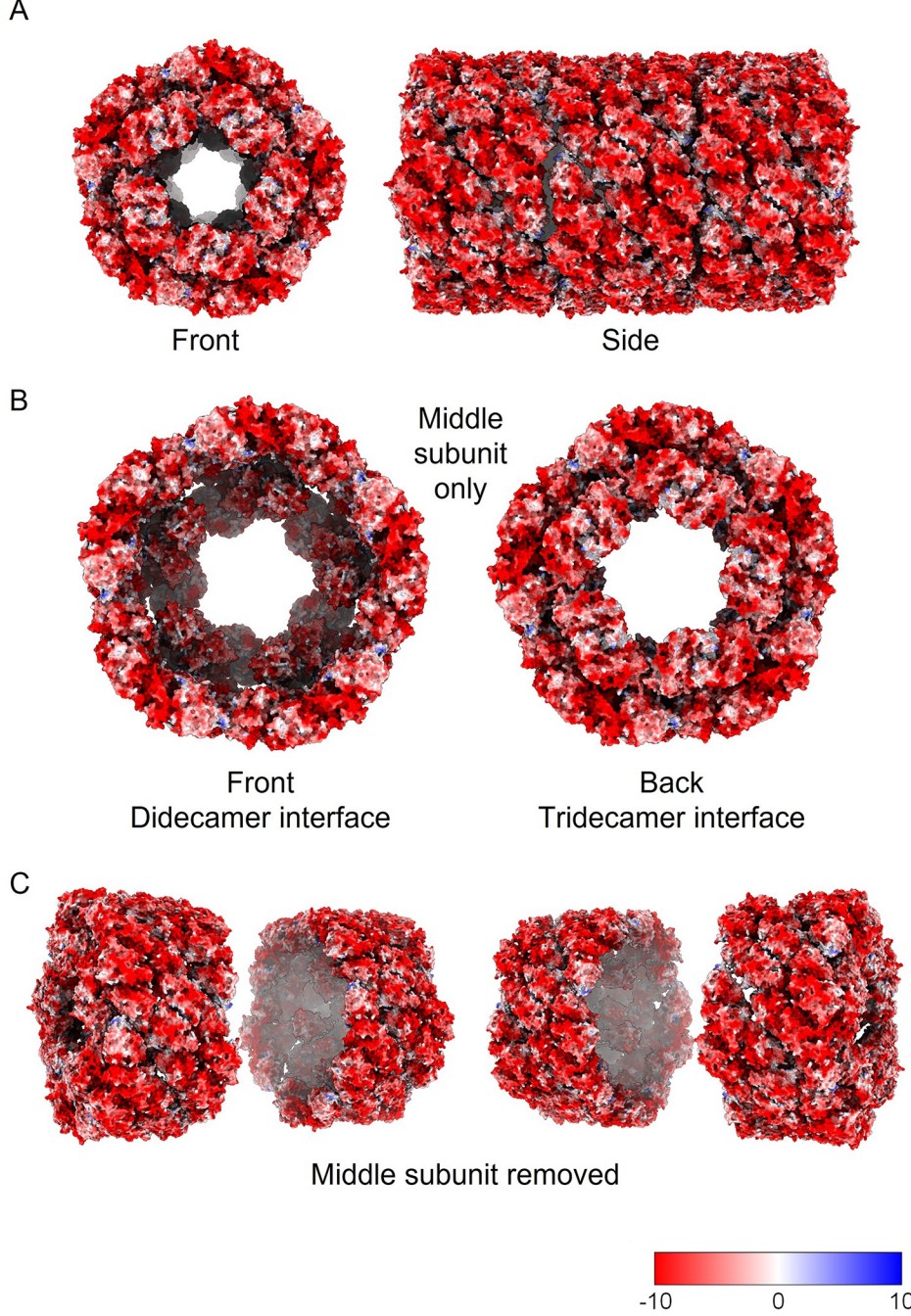

**Fig 6. Electrostatics of SLH and the arrangement of decamer building blocks.** A. Front and side views of the tridecamer model colored by electrostatics. B. Views of the front and back of the 'middle decamer' in the tridecamer structure showing the didecamer and didecamer-tridecamer interfaces. C. Oblique views of the tridecamer with the middle decamer removed. Bottom right, electrostatics scale.

heterogeneity using ion exchange chromatography or a variety of other methods including alteration of pH, buffer conditions and purification strategy. An interesting future approach would be to use size-exclusion chromatography-multi-angle light scattering (SEC-MALS) to further analyse the oligomer formation and potentially resolve the proportions of each oligomer to complement our TEM analysis.

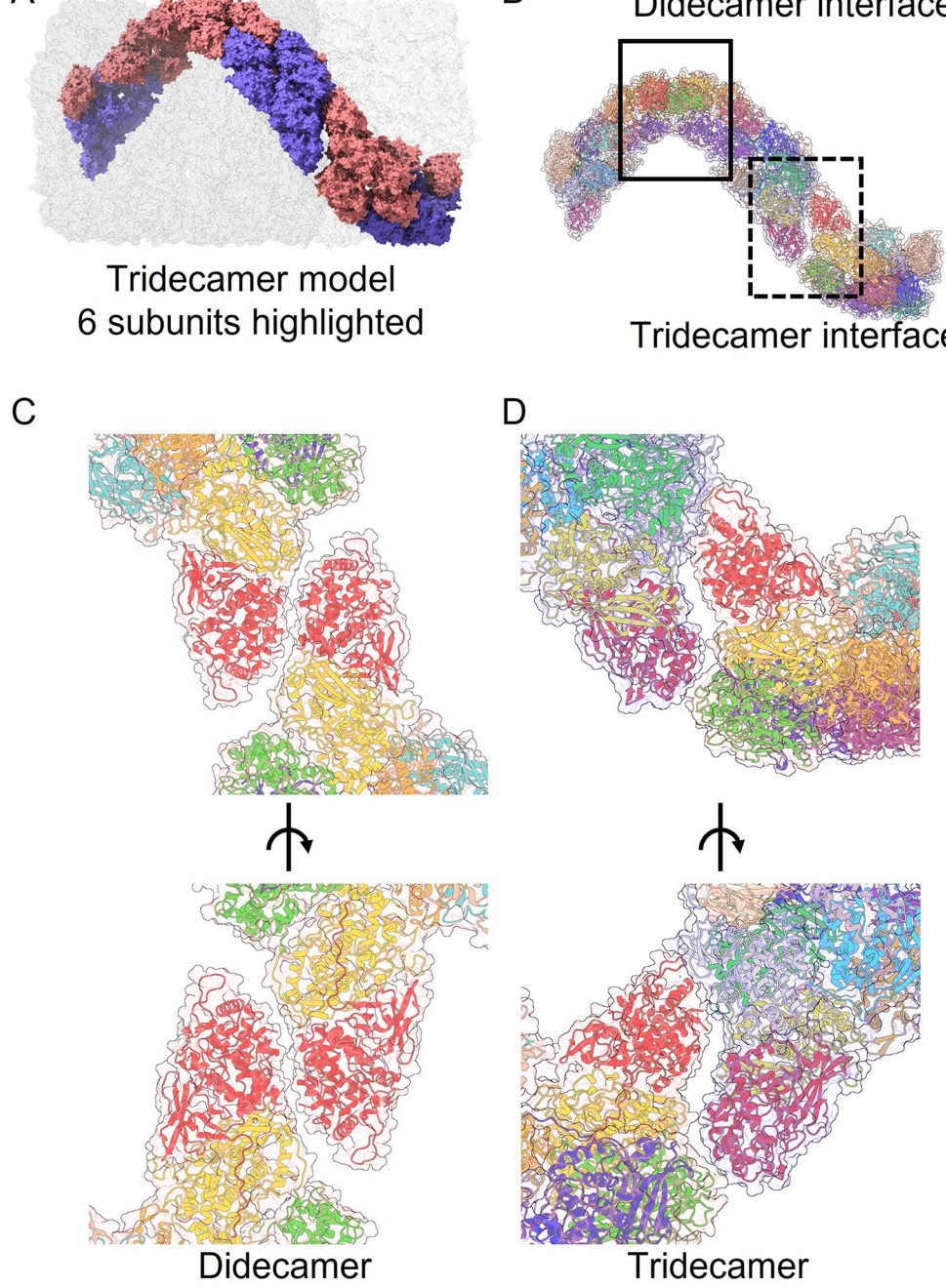

**Fig 7. Slipper limpet hemocyanin didecamer and tridecamer interface.** A. Space-filling model of the tridecamer with 6 subunits highlighted (monomer 1 pink, monomer 2 pale blue). B. Cartoon model of the 6 subunits colored by functional unit according to the key in Fig 3 highlighting the didecamer (rectangle; solid lines) and tridecamer (rectangle, dashed lines) interfaces. C. Cartoon view of the didecamer interface (bottom image rotated 180%). D. Cartoon view of the tridecamer interface (bottom image rotated 180%).

It could be that, when isolated from the hemolymph, SLH aggregates to form larger multi-mers, and the didecamer is the biologically relevant species. Alternatively, it could be that higher order oligomers are naturally present in hemolymph, and that they have a biological function. We have not performed any functional assays on SLH, but in other hemocyanins,

oxygen binding is cooperative [2], and damage to the protein by either partial denaturation or proteolysis 'relaxes' the 3D-structure of the protein, inducing tyrosinase enzyme activity [18,19]. We anticipate that SLH may also display these functional properties and determining the role (if any) of higher order oligomer formation in these processes would be an interesting avenue for future studies. Hemocyanins are known to participate in molluscan innate immune responses [20] and multimer formation may be important in this process. Alternatively, multimer formation may increase the quantity of soluble hemocyanin in the hemolymph, increasing oxygen capacity, or be part of a process of protein turnover, where larger multimers are preferentially selected for degradation.

We used two strategies when collecting sets of particles for structure determination; automated particle selection for tridecamers (using crYOLO [16] to train automated selection in Relion [15]) and manual selection for didecamers. In either case the total number of particles that we recovered was relatively low (4149 for the tridecamer and 1541 for the didecamer). The highest-resolution final structure was the tridecamer, at 4.7 Å, built using c5 symmetry, but the didecamer structure could only be improved to 7.0 Å using d5 symmetry and a subset of 710 particles. It was very clear from our tridecamer structure that both tail-tail and head-tail arrangements of decamers were possible, and so we attempted reconstructions of the didecamer particle set using c5 symmetry, but this did not improve resolution or reveal a subset of didecamers with a head-tail arrangement of decamers. It may be that a core tail-tail didecamer is the most favorable interaction, and that head-tail interactions can build mega-structures by adding decamers at either (or both) ends of the molecule. Increasing the number of particles may improve resolution in further studies, but this will require significant sample optimisation to reduce heterogeneity, or a much larger dataset. One problem with this approach is the potential to incorrectly assign tilted particles as either didecamers or tridecamers, but we reasoned that ice thickness would limit the ability for large multimers to be tilted in our samples. We struggled to obtain high-quality grids when preparing samples using traditional plunge-freezing methods, but the Chameleon automated system produced samples with excellent ice thickness, and a good distribution of particles in the grid holes.

Alphafold is an excellent system for *ab initio* modelling of protein 3D-structure, and is generally highly accurate for small, globular proteins that do not form multimers [21]. The SLH 1 sequence that we assembled from transcriptome data, at 3424 amino acids in length, was too large to model as one unit, and so we split the sequence into 6 overlapping fragments (approx size 800 amino-acids), modelling each separately and then fitting together to produce the final model. We reasoned that as hemocyanin is composed of 8 functional units, all with a similar core structure, the Alphafold model would be accurate for these sub sequences. Once constructed, there were only two ways of modelling each subunit to fit our data (one compact, and one with relatively extended linkers between FU-g and FU-h); these produced a dimer model with excellent structural homology with that of KLH (PDB 4BED; [4]). This subunit arrangement is in contrast to that proposed for *Haliotis diversicolor* hemocyanin [12], where more extensive domain swapping is observed between FU-e and FU-f. The resolution of our cryoEM structures is not sufficient to precisely confirm the arrangement of FUs by subunit; we can only state that our molecular model fits our maps well, but higher resolution structures will be needed to rule out more extensive domain swapping in the dimer plate.

Molluscan hemocyanins are secreted glycoproteins, and glycosylation can account for up to 9% of their molecular mass [22]. This glycosylation is important for their structural stability and immunomodulatory properties [23]. Analysis of the protein sequence of SLH 1 revealed 11 potential N-linked glycosylation acceptor sequences, of which 3 were conserved with KLH (Asn-410 in FU-a, Asn-1657 in FU-c and Asn-2493 in FU-f). Comparison of our molecular model with the tridecamer cryoEM structure did not immediately reveal substantial areas of

density unattributable to the protein structure (as seen for KLH1 [4]), but we were able to see this in the didecamer structure, and determine that 8 of the 11 Asn residues (the three conserved sites and Asn-471 (FU-b), Asn-965 (FU-c) Asn-1045 (FU-c), Asn-1808 (FU-e) and Asn-3312 (FU-h)) were positioned within the cryoEM density, potentially suggesting that they may be associated with non-protein density. From our data we tentatively suggest that SLH may potentially have 8 N-glycans per subunit, more than the 6 observed for KLH [4].

In summary, our work has, for the first time, elucidated the 3D-structure of slipper limpet hemocyanin, combining cryoEM with Alphafold modelling, and demonstrated how higher-order oligomers can form in hemocyanins of the KLH class.

## Methods

### Isolation and preparation of slipper limpet hemocyanin

Hemocyanin from *Crepidula fornicata* (slipper limpet hemocyanin, SLH) was isolated fresh from slipper limpet hemolymph according to Mikota's optimised protocol and stabilised in Phase 2 buffer (50 mM Tris-HCl, 150 mM NaCl, 5 mM $MgCl_2$ and 5 mM $CaCl_2$). Hemocyanin from *Megathura crenulata* (keyhole limpet hemocyanin, KLH) was obtained from Sigma-Aldrich. Bovine Serum Albumin (BSA) was obtained from Formedium.

### Sample analysis by SDS-PAGE, gel filtration and ion exchange chromatography

The protein content of hemocyanin preparations was quantified using the Bio-Rad Protein Assay reagent according to manufacturers' instructions. SDS-polyacrylamide gel electrophoresis (SDS-PAGE) was performed according to manufacturer's instructions using Nu-PAGE™ 3–8% Tris-Acetate precast gels and 1X NuPAGE™ Tris-Acetate SDS Running Buffer 20X (Thermo Scientific). HiMark™ Pre-stained Protein Standard 31–460 kDa (Thermo Scientific) was used for molecular weight estimation. Gels were stained with InstantBlue® Coomassie Protein Stain (Abcam) and scanned with an Odyssey® CLx imager (LI-COR Biosciences). Gel filtration chromatography was performed on an Akta Pure 25 instrument (GE Healthcare Life Sciences) equipped with Superose® 6 10/300 GL column (Cytiva). Samples were filtered through cellulose acetate 0.2 μm syringe filters before injection. Phase 2 buffer was used for elution unless otherwise specified. Absorbance at both 280 nm (aromatic amino acids) and 350 nm (type 3 copper centre) were monitored. Ion exchange chromatography was performed on an Akta Prime Plus instrument (GE Healthcare Life Sciences) fitted with a HiTrap™ SP FF 1 mL column according to manufacturer's instructions. SLH samples were loaded *via* a 10 mL loop after filtration through cellulose acetate 0.2 syringe filters. Absorbance was monitored at 280 nm. Samples from flow-through (manually collected) and 1 mL fractions were then resolved on SDS-Page.

### Transmission electron microscopy (TEM)

400 mesh carbon-coated copper grids (Agar Scientific) were glow discharged (25 mA for 20 seconds) before sample application. Approximately 3 μL SLH (100 ng/μL concentration) was applied to the grids for one minute. Excess liquid was blotted away using Whatman® No. 3 Filter paper before washing the grids twice with ultrapure water and blotting excess liquid. Staining was performed with 10 μL of UA-Zero® EM stain (Agar Scientific) for one minute before blotting away the excess. Grids were stored in a grid box at room temperature until imaged. Samples were imaged using a JEOL JEM-2100 transmission electron microscope operating at 200 kV.

## Cryo-EM imaging

Initially cryo-EM grids of protein purified by ion exchange chromatography (flow through fraction) were prepared at Bristol GW4 cryoTEM facility using an FEI Vitrobot. Ice quality was assessed, and initial data collected on the FEI Talos 200kV cryoTEM. Subsequently, to optimise sample preparation, the Chameleon system was used at eBIC National Facility. Purified protein was used at a final concentration of 15 mg/ml and centrifuged at 21000g, 4˚C prior to grid preparation. SPT Labtech prototype 300 mesh 1.2/0.8 nanowire grids were glow-discharged at 12 mA for 18 s. The protein was applied to the grids using a Chameleon system (SPT Labtech) at 70% relative humidity, ambient temperature and frozen in liquified ethane at 164 ms dispense-to-plunge time. Grids were screened and data collected using a Glacios microscope (Thermo Fisher Scientific). equipped with a Falcon 4 camera. 30-frame movies were collected in counting mode using EPU (Thermo Fisher Scientific) with a defocus range of -0.7 to -2.2 μm at a nominal magnification of x92,000, corresponding to a calibrated physical pixel size of 1.6 Å/pixel (5.7 e/pix/s). Total exposure time was 13 seconds, total dose was 74.1 e/pix (28.95 e/Å$^2$) and the dose per fraction was 2.47 e/pix (0.96 e/Å$^2$).

## CryoEM data processing and structure generation

Data processing and structure generation was done using Relion 3.1 [15]. Movies were processed using the Relion_IT.py pipeline, motion corrected using motioncor2 [24] in Relion (v3.1) with a 5x5 patch-based alignment and CTF-estimation was performed using Gctf (v1.18) [25]. A total of 1541 didecamer particles were manually selected; with these particles the final resolution was approx. 9 Å; using a subset of 710 particles and d5 symmetry this improved to 7 Å. For the tridecamer structure, non-template-driven particle picking was performed within crYOLO [16] using a general model, followed by reference-free 2D classification. Classes showing different orientations of hemocyanin were selected and used as templates for autopicking within Relion. The final set of 4149 particles was selected by multiple rounds of 2D classification, followed by 3D classification. The best 3D class was then refined, CTF-refined and Bayesian-polished within Relion. Additional faint density could be observed above and below the tridecamer structure in the pre-polished raw maps (presumably due to some heterogeneity remaining in the particle data set (e.g. a small proportion of tetra- or penta-decamers)), and the resolution of these maps was estimated to be 7.7 Å. Following masking, the final resolution of the tridecamer structure was measured at 4.7 Å. ChimeraX was used to display all models [26].

## Transcriptome analysis

Sequence assembly was performed using previously deposited slipper limpet transcriptome data. Bioproject PRJNA249058 (Comparative population genomics in 76 metazoan species (popphyl project) containing 8 SRA samples, SRR1324873, SRR1324874, SRR1324875, SRR1324876, SRR1324877, SRR1324878, SRR1324879 and SRR1324880) provided sequence data sufficient for analysis. The SRA accessions for *Crepidula fornicata* were concatenated and represented 39,362,017 pair end 75 bp reads. Initial quality trimming was performed yielding 97% paired reads passing QC threshold using trimmomatic (v0.36) [27]. The trimmed reads were assembled using Trinity(v2.5.0) [28] yielding 208,199 transcripts from 127,279 gene objects with a gene N50 of 632 bp. Redundant and aberrant assembly products were removed using an Evidentialgene approach (Evigene v18jan01) [29] which substantially improved the assembly producing 66,595 primary transcripts representative of 59,171 gene objects with an N50 for the longest isoform of 946 bp. The full length hemocyanin from *Aplysia californica* (Genbank accession AJ556169) was used to interrogate all transcripts encoding Hc using

NCBI-Blast+(v2.6.0) [30]. All *C. fornicata* contigs encoding Hc fragments were used as a 'bait' to identify primary reads contributing to these elements by aligning the reads passing primary QC using bbmapper tool mapPacBio [31] under default parameters defined by the slow = f flag. Samtools(v1.5) [32] was used to extract paired reads where one of the paired sequences aligned to the Hc contigs, and these reads were subsequent used to perform a *de-novo* assembly using SPades (v3.11.1) [33]. This process yielded three transcripts encoding Hc open reading frames which were validated by re-mapping the assembled transcripts using high sensitivity paired data re-mapping in Geneious (v9.1.8). The Hc1 transcript was 11,378 bp and encoded a predicted a full length Hc product of 3,432 amino acids (which we have termed SLH1); although read mapping was highly biased towards the 3'-end of the transcript, the assembly was supported by >30 fold coverage across its complete length.

## Structural modelling of SLH using Alphafold

The sequence provided was split into 6 sequences which were a maximum of 800 amino acids in length with a portion overlapping. Each sequence was modelled using the AlphaFold2 [21] implementation within AlphaFold2_advanced notebook from ColabFold [34]. This was run using Google Colaboratory computational resources using the following settings, msa_method = mmseqs2, homooligomer = 1, pair_mode = unpaired, cov = 0, qid = 0, max_msa = 512:1024, subsample_msa = True, num_relax = 0, use_turbo = True, use_ptm = True, rank_by = pLDDT, num_models = 5, num_samples = 1, num_ensemble = 1, max_recycles = 12, tol = 0, is_training = False, use_templates = False). For each modelling run, 5 structures were generated with the top predicted local-distance difference test (plDDT) ranking structures being carried forward for further processing and density fitting. The scores for each model used can be found in S1 Table. The plDDT test score is a confidence scale from 0–100 (0 = lowest confidence, 100 = highest confidence), correlates with a lDDT-Cα metric [35] and is considered an accurate estimator of structure precision.

Using ChimeraX [26], the structure of the keyhole limpet hemocyanin (4BED) dimer was fitted to the cryoEM map of SLH using "fit to map". Using "matchmaker", the predicted structures of SLH were then fitted to 4BED. Once all key domains of the predicted hemocyanin were fitted to 4BED, 4BED was removed and the Alphafold predicted structures were fitted directly to the EM map, therefore the 4BED structure was used as a guide between the predicted SLH structures and the SLH cryoEM map. The tether sequences between the domains were removed prior to fitting and then re-modelled using ModLoop [36].

## Supporting information

**S1 Fig. Ion exchange chromatography of SLH.** A. Separation of SLH isoforms by Anion Exchange Chromatography (AEC) using HiTrap Q FF 1ml column. Blue trace shows 280 nm protein signal. ~97.5 mg sample load (10 mL loop), Phase 2 buffer (pH 7.4) eluted with gradient of 1M NaCl Phase 2 buffer (green trace showing % of elution buffer). Flow-through peak (blue arrow) plateauing due to the amount of protein loaded and the flow speed of the column (1 ml/min). Eluted minor peak (fractions 9–10) shown by a red arrow. B-C. Gel filtration (Superose® 6 10/300 GL column) traces of flow-through (B) and fraction 9–10 (C) showing 280 nm protein signal (blue) and 350 nm copper-bound signal (orange). D: SDS-PAGE of Size comparison of AEC Flow-through (AEC FT) and AE Fraction 9–10 (AEC Fr 9 and AEC Fr 10, respectively) to un-separated SLH (SLH) and KLH. 1 µg of KLH and 2.5 µg SLH samples were loaded.
(JPG)

**S2 Fig. Analysis of SLH heterogeneity.** A: Representative transmission electron micrograph of 100 ng/μl flow-through and fraction 9 maintained in Phase 2 buffer negatively stained with UA-Zero. Fraction 9 was ran a total of 3 times through the AE column before the sample was used for grid preparation. Scale bar = 200 nm. B: Histograms of particle length (side-viewed particles only) and pie charts of top:side view ratio of unseparated SLH, flow-through and fraction 9 samples in TEM micrographs. 'n' indicates number of measured particles in each sample. C: Cartoon model depicting assembly of higher-order SLH monomers from a 'core' didecamer.
(JPG)

**S3 Fig. CryoEM analysis of SLH.** A. Representative micrographs (after motion correction from grids prepared using the chameleon and imaged on the FEI Glacios instrument. Significant particle heterogeneity is observed. B-C. 2D-class averages from the didecamer (B) and tridecamer (C) reconstructions. D-E. FSC plots from the didecamer (D) and tridecamer (E) reconstructions (dashed line represents FSC = 0.143), with final resolutions indicated.
(JPG)

**S4 Fig. Inferred SLH1 protein sequence.** Sequence represents the longest SLH open reading frame (here termed SLH1; derived from publicly deposited transcriptome data) colored by functional unit according to the key in Fig 3 (apart from FU-c (yellow background with bold text) and FU-h (grey background). SLH1 shares 55.5% identity at the amino acid level with KLH1 (Q10583) and 54.9% identity with KLH2 (Q10584).
(JPG)

**S5 Fig. Analysis of potential SLH N-glycosylation.** A. NetNGlyc output showing predicted N-glycosylation acceptor sequences for SLH. Sequences conserved between SLH and KLH are highlighted in black. The right-hand column states whether the asparagine side-chain from the alphafold tridecamer model is exposed (No) or buried (Yes) in the cryoEM map; those labelled 'yes' may be glycosylated in mature SLH. B. Front and side views of the SLH tridecamer cryoEM structure (white) fitted to the molecular model with conserved (red) and non-conserved (blue) asparagines from potential N-glycosylation sequences shown as sheres. C. Close-up front and side views of the tridecamer structures with asparagine residues labelled; for some, the Asn sidechain is buried within the cryoEM volume (pale red or pale blue), whereas for others the sidechain appears exposed. D. Views of the didecamer cryoEM structure (pink) with the SLH molecular model (grey ribbons) fitted, with potentially glycosylated Asn residues shown as spheres. Close-up views showing the additional density surrounding Asn-1657 and Asn-1045 are also shown.
(JPG)

**S1 Table. plDDT values for the 6 SLH initial Alphafold models.**
(PDF)

**S1 Raw images.**
(PDF)

# Acknowledgments

The authors would like to thank the Biomolecular and Bioimaging Technology Hubs (School of Biosciences, Cardiff University) for assistance with protein analysis and image preparation, the GW4 Facility for High-Resolution Electron Cryo-Microscopy (University of Bristol) and the Electron Bio-Imaging Centre (Diamond Light Source) for assistance with sample

preparation and cryoEM imaging, and the Advanced Research Computing at Cardiff (ARCCA; Cardiff University) for assistance with data processing.

## Author Contributions

**Conceptualization:** Gaia Pasqualetto, Andrew Mack, Emily Lewis, Alex Muhlhozl, Mark T. Young.

**Data curation:** Ufuk Borucu, Tom Davies, Miriam Weckener, Dan Clare, Tom Green, Mark T. Young.

**Funding acquisition:** Alex Muhlhozl, Mark T. Young.

**Investigation:** Gaia Pasqualetto, Andrew Mack, Emily Lewis, Ryan Cooper, Alistair Holland, Ufuk Borucu, Judith Mantell, Tom Davies, Miriam Weckener, Dan Clare, Pete Kille, Mark T. Young.

**Methodology:** Gaia Pasqualetto, Andrew Mack, Emily Lewis, Ryan Cooper, Alistair Holland, Ufuk Borucu, Judith Mantell, Tom Davies, Miriam Weckener, Dan Clare, Pete Kille, Alex Muhlhozl.

**Project administration:** Alex Muhlhozl, Mark T. Young.

**Resources:** Alex Muhlhozl.

**Software:** Dan Clare, Tom Green.

**Supervision:** Alex Muhlhozl, Mark T. Young.

**Validation:** Tom Green.

**Writing – original draft:** Mark T. Young.

**Writing – review & editing:** Gaia Pasqualetto, Andrew Mack, Ufuk Borucu, Judith Mantell, Miriam Weckener, Pete Kille, Mark T. Young.

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
