## [Decision Letter · Decision Letter 0]

11 Apr 2023

PONE-D-23-02379CryoEM structure and Alphafold molecular modelling of a novel molluscan hemocyaninPLOS ONE

Dear Dr. Young,

Thank you for submitting your manuscript to PLOS ONE. After careful consideration, we feel that it has merit but does not fully meet PLOS ONE’s publication criteria as it currently stands. Therefore, we invite you to submit a revised version of the manuscript that addresses the points raised during the review process. As you can see below both Reviewers positively evaluated your manuscript, which in my opinion could lie at the interface between a methodological work clearly focused on structure and a biological research paper. For the latter, one feels that some information are missed.

Indeed, Reviewer #2 correctly suggested the addition of mutational and functional studies could help a better characterization of this protein improving the quality of the work.

As pointed out even by this Reviewer (e.g. *can explain.., can help..*), I believe in this case the new experimental data are not strictly necessary but more recommended. Instead, a deeper structural characterization/discussion of the diverse interfaces responsible for the oligomerization in the result section aligned with an appropriate discussion is required to top up the gap mentioned above.

We look forward to receiving your revised manuscript.

Kind regards,

Matteo De March

Academic Editor

PLOS ONE

Journal Requirements:

Reviewers' comments:

Reviewer's Responses to Questions

**Comments to the Author**

1. Is the manuscript technically sound, and do the data support the conclusions?

Reviewer #1: Yes

Reviewer #2: Yes

2. Has the statistical analysis been performed appropriately and rigorously? 

Reviewer #1: Yes

Reviewer #2: Yes

3. Have the authors made all data underlying the findings in their manuscript fully available?

Reviewer #1: Yes

Reviewer #2: Yes

4. Is the manuscript presented in an intelligible fashion and written in standard English?

Reviewer #1: Yes

Reviewer #2: Yes

5. Review Comments to the Author

Reviewer #1: Authors of this paper describe the cryoEM structure of the didecamer and tridecamer froms of a novel hemocyanin from the SLH, and use Alphafold to build large oligomeric proteins. However, I have some questions:

1. In SI page 4, line 11, Fig S4, some of them are identified by the color of the font itself, while others are identified by background highlighting，please use the same identification method to identidy functional units.

2. In paper page 18, line 395-396, "The sequence provided was split into 6 sequences which were a maximum of 800 amino acid in length with a portion overlapping". What is the basis for the truncation except for the maximum limit of Alphafold?

Reviewer #2: The work of science is highly interesting which elucidates the 3D-structure of slipper limpet hemocyanin, combining cryoEM with Alphafold modelling, and demonstrated how higher-order oligomers can form in hemocyanins of the KLH class. I have some questions which I feel can help in the development of the work.

1. The authors have explained that they can see higher order oligomers in the sample produced. I would like to know which part of the protein helps in this oligomerisation. SEC-MALS studies as well as mutational studies ca explain the oligomerisation formation.

2. How important s it for hemocyanin to form oligomer for its functionality? Any in-vitro experiment elucidating the function can help in the understanding of the formation of higher order oligomers.

3. A cartoon model of the formation of oligomer from monomer can help in better understanding of the work.

6. PLOS authors have the option to publish the peer review history of their article (what does this mean?). If published, this will include your full peer review and any attached files.

Reviewer #1: **Yes: **Xiaomin Ma

Reviewer #2: No

---

## [Author Response · Author response to Decision Letter 0]

24 May 2023

Reviewer 1

Authors of this paper describe the cryoEM structure of the didecamer and tridecamer froms of a novel hemocyanin from the SLH, and use Alphafold to build large oligomeric proteins. However, I have some questions:

1. In SI page 4, line 11, Fig S4, some of them are identified by the color of the font itself, while others are identified by background highlighting，please use the same identification method to identidy functional units.

We have used the background highlighting for the functional units to keep this consistent.

2. In paper page 18, line 395-396, "The sequence provided was split into 6 sequences which were a maximum of 800 amino acid in length with a portion overlapping". What is the basis for the truncation except for the maximum limit of Alphafold?

The basis for this truncation was indeed the maximum limit for Alphafold.

Reviewer 2

The work of science is highly interesting which elucidates the 3D-structure of slipper limpet hemocyanin, combining cryoEM with Alphafold modelling, and demonstrated how higher-order oligomers can form in hemocyanins of the KLH class. I have some questions which I feel can help in the development of the work.

1. The authors have explained that they can see higher order oligomers in the sample produced. I would like to know which part of the protein helps in this oligomerisation. SEC-MALS studies as well as mutational studies ca explain the oligomerisation formation.

This is a very interesting suggestion. We looked at using SEC-RALS to look at oligomer formation, but we were not able to resolve single peaks for any species in our SEC experiments and so we did not continue with these experiments. From visual inspection of our negative-stain and cryoEM images we always observe the core didecamer (a tail-tail interaction) in any oligomer, and then additional subunits add on in a head-tail interaction. From our fitting of the Alphafold models to the tridecamer structure, we observe a small difference in the subunit interface in the head-tail interaction compared to the tail-tail interaction. It may be that in other hemocyanins that do not show substantial higher-order oligomer formation, this interaction is prevented or disfavoured by glycosylation or simply lack of an interface. Mutational studies are an excellent suggestion to resolve this, but we are working with protein purified from a native source in an organism that is not amenable to cloning for site-directed mutagenesis, so it is not possible for us to address this point. We have included some additional information in the discussion to cover this point (page 12).

2. How important s it for hemocyanin to form oligomer for its functionality? Any in-vitro experiment elucidating the function can help in the understanding of the formation of higher order oligomers.

This is a very interesting point – unfortunately we do not know about the biological relevance of the higher order oligomers, or indeed the oxygen binding or enzyme function of our SLH didecamer, but from comparison with other hemocyanins, we know that oxygen binding is cooperative (so subunit-subunit interactions are critical for this process), and that partial denaturation or proteolysis produces a functional oxidase enzyme with a ‘relaxed’ structure relative to the intact protein. Functional experiments are something that we would very much like to address in future experiments, but we think they are beyond the scope of the structural and modelling experiments that we present in this manuscript. We have included some additional information in the text (pages 12 and 13) to discuss these points.

3. A cartoon model of the formation of oligomer from monomer can help in better understanding of the work.

We agree that this information would be very useful. We do not have any experimental evidence for how a decamer might form from a monomer (we assume a monomer forms a dimer ‘plate’ and then 5 ‘plates assemble to the didecamer). We have included a simple cartoon model describing how decamers may join together to assemble didecamers and higher-order structures (Fig S2C).

---

## [Decision Letter · Decision Letter 1]

5 Jun 2023

CryoEM structure and Alphafold molecular modelling of a novel molluscan hemocyanin

PONE-D-23-02379R1

Dear Prof. Mark Young,

We’re pleased to inform you that your manuscript has been judged scientifically suitable for publication and will be formally accepted for publication once it meets all outstanding technical requirements.

Kind regards,

Matteo De March

Academic Editor

PLOS ONE

Additional Editor Comments (optional):

Reviewers' comments:

Reviewer's Responses to Questions

**Comments to the Author**

1. If the authors have adequately addressed your comments raised in a previous round of review and you feel that this manuscript is now acceptable for publication, you may indicate that here to bypass the “Comments to the Author” section, enter your conflict of interest statement in the “Confidential to Editor” section, and submit your "Accept" recommendation.

Reviewer #1: All comments have been addressed

Reviewer #2: All comments have been addressed

2. Is the manuscript technically sound, and do the data support the conclusions?

Reviewer #1: Yes

Reviewer #2: Yes

3. Has the statistical analysis been performed appropriately and rigorously? 

Reviewer #1: Yes

Reviewer #2: Yes

4. Have the authors made all data underlying the findings in their manuscript fully available?

Reviewer #1: Yes

Reviewer #2: Yes

5. Is the manuscript presented in an intelligible fashion and written in standard English?

Reviewer #1: Yes

Reviewer #2: Yes

6. Review Comments to the Author

Reviewer #1: Authors of this paper describe the cryoEM structure of the didecamer and tridecamer froms of a novel hemocyanin from the SLH, and use Alphafold to build large oligomeric proteins. And they answered my questions. I agree to publish this paper.

Reviewer #2: The work of science need to accepted with no further delay. The reviews are well addressed by the authors. Further functional analysis in future will be a great addition.

7. PLOS authors have the option to publish the peer review history of their article (what does this mean?). If published, this will include your full peer review and any attached files.

Reviewer #1: **Yes: **Xiaomin Ma

Reviewer #2: No

---

## [Editor Report · Acceptance letter]

13 Jun 2023

PONE-D-23-02379R1 

CryoEM structure and Alphafold molecular modelling of a novel molluscan hemocyanin 

Dear Dr. Young:

I'm pleased to inform you that your manuscript has been deemed suitable for publication in PLOS ONE. Congratulations! Your manuscript is now with our production department. 

Kind regards, 

on behalf of

Dr. Matteo De March 

Academic Editor

PLOS ONE